# Synthesis and Advanced NMR Characterization of Ordered 3D Reticular Materials with PolySilicate Nodes and Hydrophobic OrganoSilicone Linkers

**DOI:** 10.3390/molecules30020228

**Published:** 2025-01-08

**Authors:** Jelle Jamoul, Sambhu Radhakrishnan, Maarten Houlleberghs, C. Vinod Chandran, Aline Vits, Pasquinel Weckx, Sam Smet, Daniel Arenas Esteban, Sara Bals, Johan A. Martens, Eric Breynaert

**Affiliations:** 1Centre for Surface Chemistry and Catalysis—Characterization and Application Team (COK-KAT), KU Leuven, Celestijnenlaan 200F Box 2461, 3001 Heverlee, Belgium; 2NMRCoRe—NMR/X-Ray Platform for Convergence Research, KU Leuven, Celestijnenlaan 200F Box 2461, 3001 Heverlee, Belgium; 3Electron Microscopy for Materials Science (EMAT), University of Antwerp, Groenenborgerlaan 171, 2020 Antwerp, Belgium

**Keywords:** reticular material synthesis, silicate-silicone hybrids, POSiSils, cyclosilicate hydrate, siloxane-silsesquioxane, octyl-methyldichlorosilane, dicyclopentyldichlorosilane, advanced multidimensional NMR characterization, ^29^Si NMR

## Abstract

This work describes the synthesis of ordered 3D siloxane-silsesquioxane reticular materials with silicate D4R cubes (Si_8_O_20_^8−^), harvested from a sacrificial tetrabutylammonium cyclosilicate hydrate (TBA-CySH) precursor, interlinked with octyl and dicyclopentyl (Cp_2_) hydrocarbon functionalities in a one-step synthesis with organodichlorosilanes. Advanced solid-state NMR spectroscopy allowed us to unravel the molecular order of the nodes and their interconnection by the silicone linkers. In the case of octyl-methyl silicone linkers, changing the silane-to-silicate ratio in the synthesis allowed for tuning the length of the linker between the nodes. With dicyclopentyl linkers, the addition of dimethyldichlorosilane was essential to enable the formation of a reticular network. The resulting materials contained mixed, dimeric silicone linkers, i.e., Si_8_-O-Si(Me_2_)-O-Si(Cp_2_)-O-Si_8_.

## 1. Introduction

Siloxane-silsesquioxane resins incorporating cubic oligomeric silicate cage units (T_8_ or Q_8_) are very popular for constructing covalently linked hybrid porous polymers with tunable structures and multifunctional properties. The cubic symmetry of the Si_8_O_12_ double 4-ring (D4R), its intrinsic structural rigidity, and nanosized dimensions (1.2–1.4 nm) render it an attractive supramolecular unit. Applications of hybrid polymers incorporating D4R cage organosiloxanes include adsorption and storage of gasses, catalysis, water treatment, drug delivery, and sensing applications. Used as nanofillers of organic polymers, they serve to enhance the thermal, mechanical, and electrical properties of the final material [1,2,3,4,5,6,7,8,9,10].

Depending on their peripheral chemical functionalities, D4R silicate units are identified under different names (Figure 1). D4Rs terminated with organic groups ([R_8_-Si_8_O_12_] units with R = H, alkyl, aryl, etc.) are typically referred to as polyhedral oligomeric silsesquioxanes (POSS). When R is O^−^ or OH, they are identified as cyclosilicates, and if R is a siloxy group, they are called spherosilicates. POSS D4R units are typically synthesized by hydrolyzing and condensing alkoxy- or chlorosilanes. This results in inefficient synthesis conditions and requires the implementation of lengthy purification to eliminate silicate byproducts, especially when small R-groups are targeted [11,12,13]. Cyclosilicate D4Rs can be harvested from cyclosilicate hydrates (CySHs). CySHs are templated hydrate crystals comprising silicate building blocks, i.e., D3R, D4R, and D6R units, next to template molecules and water [14,15,16,17]. Spherosilicate D4Rs can be obtained either from POSS or from cyclosilicate hydrates [13,17,18], with the latter route circumventing the drawbacks of POSS syntheses [7,8,17,19]. When synthesizing spherosilicates from CySH precursor crystals, the template used is extremely impactful as it determines the crystal structure and, consequently, the amount of crystal water incorporated. Limited water contents are preferred as they enable the selective conversion of cyclosilicates into spherosilicate units using chloro- or dichlorosilanes, which otherwise react with superfluous water to form silicone polymers rather than spherosilicates [7].

POSS D4Rs can be coupled into hybrid materials using a multitude of linkers. Purely organic linkages can be created by metal-catalyzed cross-coupling reactions (Sonogashiri and Friedel-Craft condensations), examples ranging from large aromatic to simple aliphatic hydrocarbons (ethyl). Such syntheses typically start from octahydrido- and octavinyl POSS units (R = H of C_2_H_3_, respectively) or their functionalized equivalents [10,20,21]. Using diethylhydroxylamine as a catalyst, H-POSS or spherosilicate equivalent units can be interlinked with silanol-terminated silanes (OH-(SiR_2_-O)_n_-H with R = phenyl and methyl) by dehydrogenative condensation [9,22]. This yields materials with pure silicone linkers. Using organic diols, the same catalyst also enables the generation of Si-O-C bonds, yielding materials with purely organic linkers [4,23].

CySHs can be obtained with a multitude of quaternary ammonium species with at least one methyl substituent (e.g., tetramethylammonium (TMA), choline, 1,1-dimethylpiperidinium, phenyltrimethyl- and benzyltrimethylammonium, piperazinium derivatives, etc.), tetrabutylammonium (TBA), and hexamethyleneimine (HMI) [14,17,19,24,25,26]. Using TMA as a template, TMA-CySH nucleates as soon as Q^3^_8_ levels reach a threshold of ≈32 % of the total silicate species [27]. Further growth yields TMA_8_Si_8_O_20_.65H_2_O, incorporating 65 water molecules per D4R in the crystal structure. This high water content negatively affects the use of TMA-CySH for functionalization with chlorosilanes due to water-induced silane polymerization, resulting in lowered synthesis efficiency and uncontrollable network propagation [26]. HMI-CySH contains 12 water molecules per cube and can successfully be reacted with dimethyldichlorosilane to yield a PolyOligoSiloxySilicone (POSiSil) material featuring monomeric dimethylsilicone-bridged silicate cubes (PSS-1) [7]. Using TBA-CySH, with only 5.33 water molecules per D4R unit, selective reaction with dimethyldichlorosilane converts the D4R units into octakis(dimethylsilyl chloride) spherosilicates. The resulting terminal Si-Cl bonds, render these spherosilicates ideal as building units for further reaction [17]. Self-polymerization of this chlorinated precursor block in the presence of traces of water yields PSS-2, a 3D POSiSil material [8]. The reaction of the chlorinated precursor with aliphatic alkanediols allows for the synthesis of a isoreticular family of PolySilicate Porous Organic Polymers (PSiPOPs), featuring Si-O-C connections between the polysilicate node and the linkers [28].

Hitherto, POSiSil materials only featured small (di)methyl silicone bridges. Consequently, in applications, they were no match for POSS-based hybrid siloxane-silsesquioxane resins incorporating long-chain organic groups. The literature has shown that such hydrocarbon functionalities enable the adsorption of aromatic solutes, such as naproxen and bisphenol, from various media, including water and milk samples, and enhance the conductivity of Li^+^ ions in solid-state battery applications [29,30,31,32]. When coordinated with metal clusters, these siloxane-silsesquioxane hybrid materials function as catalysts for C-C cross-coupling reactions, facilitating the synthesis of complex and fine chemicals [33,34].

Recently, POSiSil-like octyl-methyl functionalized silicate-silicone hybrids were synthesized by reacting TMA-CySH with dichloromethylsilane ((CH_3_)Si(H)Cl_2_). Grafting of the Si-H groups in the resulting spherosilicate with 1-octene yielded octyl-functionalized silicone linkers [35]. Structurally incorporating similar siloxane-silsesquioxane resins into polypropylene polymer blends was shown to improve their flame retardancy [36]. However, the unfavorable water content of TMA-CySH renders this synthesis route highly inefficient with respect to Si [35,37,38,39,40]. The present work presents new silicate-silicone hybrid materials with methyloctyldisiloxane and mixed dimethyl-dicyclopentyl disiloxane linkers, synthesized in an efficient manner based on the reaction of TBA-CySH with long-chain and bulky dichlorosilanes. Rather than anchoring octyl and cyclopentyl groups on siloxane linkages by post-synthetic grafting of a spherosilicate, the new synthesis strategy directly incorporates the bulky organic groups during the initial reaction with cyclosilicate hydrate, while also avoiding extensive generation of side products. Implementing this strategy with octylmethyldichlorosilane or a mixture of dimethyl- and dicyclopentyldichlorosilanes yielded reticular materials coined PSS-3 and PSS-4, respectively (Figure 2). These new POSiSil materials were characterized extensively with quantitative ^1^H and ^29^Si NMR spectroscopy, TGA, SEM, and HAADF-STEM.

## 2. Results and Discussion

### 2.1. Synthesis and Structure Elucidation of PSS-3

Octyl-methyl POSiSil samples (PSS-3) were synthesized using different concentrations of n-octyl(methyl)dichlorosilane, by adding liquid silane to vacuum-dried TBA-CySH suspended in dry THF. Based on the PSS-2 synthesis protocol, 10 silane molecules are needed per silicate cube for the complete conversion of CySH to PSS. Hereafter, this ratio is referred to as a stoichiometric amount of silane [8]. Three concentrations, i.e., 50% deficit, stoichiometric, and 50% excess of n-octyl(methyl)dichlorosilane were added. After evaporation of the high-boiling silane and THF at 90 °C and vacuum, intermediate spherosilicates were resuspended in THF and converted to octyl-methyl POSiSils by hydrolysis with water vapor. X-ray diffraction of PSS-3 identified the material as amorphous (Appendix A). As the reaction of chlorosilanes with silicate cubes of CySH in THF progresses, the hydrogen bonding network of the initial CySH crystals is disrupted. This liberates individual silicate cubes from the LTA-like topology of the starting CySH crystals [17]. While the material is reticular, the intrinsic flexibility of the silicone linkers bridging the silicate cubes renders the final POSiSil materials amorphous [8]. Consequently, NMR spectroscopy is required to elucidate the local chemical order and structure of PSS-3 (Figure 3). Previous reports on silicate-silicone hybrid materials with octyl-methyl functionalities synthesized from TMA-CySH, limited their characterization to qualitative trends observed with infrared spectroscopy and direct excitation ^29^Si NMR. The length of silicone linkers was estimated from the initial silane concentration, but given the high water content of TMA-CySH, promoting random polymerization of the silane, a more accurate quantification of the linker length is recommended [35,37].

^29^Si NMR spectroscopy allows for differentiation between Si atoms of silicate cubes (Q^3^ and Q^4^) and silicone linkers (D^2^), resonating around −100 to −110 ppm and between −15 and −23 ppm, respectively (Figure 3a). The relative concentrations of these different ^29^Si species can be derived from the spectral decomposition of the ^29^Si NMR spectra (Appendix A) [41,42]. The results are summarized in Figure 3e. Efficient silylation of the silicate cubes with octyl-methyl silicone moieties is confirmed by the strong Q^4^ Si signal (Figure 3d). Since Si D4R cubes cannot directly fuse, the Q^4^/Q^3^ ratio represents an efficiency factor for the conversion of Q^3^ Si in CySH to Q^4^ Si by reaction with silane. Based on the structure of PSS-2, exhibiting disiloxane linkers between the D4Rs, adding a 50% deficit of silane theoretically can only functionalize five out of eight corners. An 87.5% conversion was nevertheless observed, implying that 7 out of 8 corners were functionalized. This implies that in contrast with the synthesis of PSS-2, where dichlorosilane was added in the gas phase, the present synthesis route using dichlorosilane in THF also allows for the linking of D4R units using a monomeric siloxane bridge.

Silyl chloride groups attached to a D4R can thus react with a free silanol of another silicate cube. Only when monomeric silicone linkers occupy two corners of different silicate cubes, can the observed degree of conversion be achieved under silane-deficient conditions. Dimeric silicone linkers, in contrast, are formed when the terminal Si-Cl bonds of two functionalized silicate cubes are hydrolyzed and condensed.

Examining the D^2^ Si atoms, multiple signals are observed in the −15 to −23 ppm range of the 1D ^29^Si NMR spectra (Figure 3b and Appendix A). These signals indicate the presence of silicone species bridging the silicate D4R cubes, as well as a polymer phase formed due to the self-polymerization of dichlorosilane upon reaction with water, introduced in the synthesis as crystal water in the CySH precursor [7,43].

^29^Si NMR resonances of D^2^ Si species from pure octyl-methyl silicone polymers (OM polymers) were identified using the 1D ^29^Si NMR spectra of such a phase, obtained by self-polymerization of n-octyl(methyl)dichlorosilane in the presence of water (Figure 3b). The OM polymer exhibited signals around −20.3 ppm and in the range of −22 to −23 ppm. The narrow signals suggest that the polymer species are relatively mobile. Signals assigned to the POSiSil phase appear between −15 and −19 ppm and at −21.6 ppm, and are also broader. The incorporation of the octyl-methyl silicone linkers between the rigid silicate cubes reduces the mobility of the silicone linkers, causing broader signals. Based on these assignments, the D^2^/Q^4^ ratio in the PSS-3 siloxane-silsesquioxane phase is 0.69 for the sample synthesized with a 50% deficit silane. This implies that monomeric and dimeric silicone linkers are present in a ratio of 4 to 3. The length of silicone linkers can also be rationalized from the chemical shift of D^2^ Si atoms. In structurally related PSS-2 materials, monomeric and dimeric linkers resonated at −14.5 and −16.5 ppm, and −18.7 ppm and −19.7 ppm, respectively. D^2^ Si atoms shift to more negative chemical shift values with increasing length of the silicone linkers [44].

The D^2^/Q^4^ ratio increases to 1 with stoichiometric amounts of silane or higher, corresponding to dimeric silicone linkers with D^2^ Si atoms resonating at −18.6 and −19.7 ppm. The same ratio would also be obtained by the presence of equal fractions of monomeric and trimeric silicone linkers. Since only a minor fraction of Si atoms exhibited a chemical shift at −21.6 ppm, i.e., more negative than those attributed to dimeric linkers, the presence of equal fractions of monomeric and trimeric linkers can be excluded. Based on the assignment of these chemical shift ranges to D^2^ Si atoms present in, respectively, monomeric, dimeric, and trimeric linkers, the maximum of 10% of the D^2^ Si atoms in the PSS phase is present in trimeric linkers. The length variation of silicone linkers also explains the presence of multiple Q^4^ Si signals of the silicate cubes in the ^29^Si NMR spectrum, ranging from −108.4 to −109.8 ppm [45,46].

When the silane concentration beyond the stoichiometric ratio is increased, the D^2^/Q^4^ ratio no longer increases. This can be explained as follows. The CySH crystal liberates individual silicate cubes in an acidified solvent (THF). The acidification results from the release of HCl upon the reaction of dichlorosilane with hydroxyl groups on the corners of the silicate D4R. TBA-CySH crystals readily dissolve in acidified THF [17]. This releases silicate D4R into the solution, where they immediately react with octylmethyldichlorosilane to form octakis(n-octylmethylsilyl chloride) spherosilicates (Figure 3c). Since the silane is added all at once in liquid form, the disintegration of the CySH crystal structure occurs fast, and octakis(n-octylmethylsilyl chloride) spherosilicates are formed simultaneously. Increasing the silane concentration beyond the stoichiometric amount, the D^2^/Q^4^ ratio of the system remained constant. The transition of spherosilicates to POSiSils involves a condensation reaction with water (vapor) and two peripheral Si-Cl bonds from the n-octylmethyl silyl moieties.

### 2.2. Synthesis and Structure Elucidation of PSS-4

Dicyclopentyldichlorosilane is used in the production of silicones with improved thermal and chemical resistance, rendering it an obvious choice for incorporation into siloxane-silsesquioxane hybrids. Replacing octyl-methyldichlorosilane with dicyclopentyldichlorosilane in the deficit synthesis protocol for PSS-3 did, however, not allow the formation of a fully connected POSiSil material. Exclusively using dichlorodicyclopentylsilane yielded a material exhibiting high fractions of defects. They are observed as silanols at the non-linked corners of the silicate cube, Q^3^ Si, and interrupted silicone linkers, (sharp) D^1^ Si atoms, appearing, respectively, at −100 and −10 ppm in the ^29^Si NMR spectrum of the material (Appendix A). The highly defective nature of the material was attributed to the steric hindrance induced by the cyclopentyl moieties, preventing the efficient formation of silicone linkers between silicate cubes. To achieve sufficient silicone network propagation, along with the incorporation of dicyclopentyl functionalities (Cp_2_), dicyclopentyldichlorosilane was used in combination with dimethyldichlorosilane in a ratio of 8 to 3. As shown by the increase of D^2^ Si atoms (-Si-O-**Si**(R_2_)-O-Si-) in the 1D ^29^Si NMR spectrum (Figure 4c), this facilitated silicone bond formation between spherosilicate units [7,8]. The 1D ^29^Si NMR further revealed mainly Q^4^ Si (Q^4^:Q^3^ = 7.7:0.3), demonstrating an efficient functionalization of the silicate D4R. Multiple D^2^ Si species were observed in the chemical shift range between −16 and −25 ppm (i.e., −16, −19.1, −21.9, and −24.8 ppm; D^2^:Q^4^ = 10.2:7.7). The relative concentrations of these ^29^Si resonances, derived by decomposing the 1D ^29^Si MAS NMR spectrum (Appendix A), are provided in the Appendix A. Since the previously reported PSS-2, incorporating exclusively dimethylsilicone linkers, only showed D^2 29^Si resonances between −16 and −21 ppm, the resonance at −24.8 ppm was assigned to a D^2^ Si atom of a dicyclopentylsilicone moiety (-Si-O-**Si**(Cp)_2_-O-Si-). The assignment of the resonance at −24.8 ppm as Si(Cp)_2_ was confirmed from the decomposition of the 1D ^29^Si and ^1^H MAS NMR spectra (Appendix A). In the ^1^H-^29^Si HETCOR spectrum, both the cyclopentyl protons and the methyl protons correlate with the Q^4^ Si atoms of the silicate D4R, thus demonstrating the successful incorporation of both silanes into the PSS-4 structure (Figure 4e). Further structure elucidation was based on ^1^H-^1^H double quantum-single quantum (DQ-SQ) MAS NMR spectroscopy, showing self-correlation of all protons in the methyl and cyclopentyl groups, consistent with their chemical structure and in agreement with the ^1^H-^13^C HETCOR (Figure 4b,d). DQ cross-correlations between methyl and cyclopentyl protons reveal their close proximity and are consistent with the formation of mixed silicone linkers, i.e., -O-Si(Me_2_)-O-Si(Cp_2_)-O-. This is further confirmed by ^1^H-^29^Si HETCOR, showing cyclopentyl proton correlations with D^2^ Si(Me_2_) atoms (Figure 4b,e,f). Overall, this confirms the assignment of the ^29^Si resonances at −16 and −24.8 ppm to, respectively, Si(Me_2_) and Si(Cp_2_) contained in the mixed silicone linkers. The resonances at −19.1 and −21.9 ppm are ascribed to outer and central Si atoms in trimeric dimethylsilicone linkers, respectively (Figure 4c,f). The formation of such trimeric dimethylsilicone linkers implies partial oligomerization of dimethyldichlorosilane via the reaction of the silane with crystal water from the TBA-CySH precursor.

Overall, PSS-4 contains a combination of dimeric, mixed Me_2_-Cp_2_ silicone linkers, and trimeric dimethylsilicone linkers, present in equal amounts. This results in a D^2^/Q^4^ ratio of 1.33, similar to what was observed for PSS-2, synthesized from the same TBA-CySH precursor with exclusive dimethyldichlorosilane to generate the linkers [8].

### 2.3. Thermal Stability of PSS-3 and PSS-4

The thermal stability of these new PSS materials was probed with thermogravimetric analysis in an inert (N_2_) atmosphere after the selective removal of TBA-Cl by initial calcination at 250 (PSS-3) and 200 °C (PSS-4) in an inert atmosphere (Appendix A). From previous literature reports, it is known that the TBA template, as a chloride salt, decomposes around 190 °C [8,28,47]. This is observed in both PSS-3 and PSS-4 (Appendix A). For PSS-4, the calcination temperature was lowered to 200 °C due to the lower thermal stability of the cyclopentyl functional groups (Appendix A).

The PSS-3 material remains thermally stable up to 400 °C, as indicated by its T_5%_. From that temperature onward, the organic octyl and methyl functionalities start to decompose, consistent with what has previously been reported for similar siloxane-silsesquioxane resins [36]. The decomposition proceeds via a multistage weight loss, only visible after fitting the DTG curve, with a main decomposition peak at 482 °C (Appendix A). The weight loss observed upon decomposition of the organic functional groups (45.4 wt%) is consistent with the composition of the silicone linkers derived from NMR spectroscopy. The theoretical weight fraction of the octyl and methyl groups, as calculated from the chemical composition (Figure 3e), is 46.5 wt%.

For PSS-4, T_5%_ is 255 °C. The DTG curve shows a multistage decomposition of the dicyclopentyl and dimethyl functional groups, with maxima of decomposition peaks occurring at 354 °C, 460 °C, 514 °C, and 670 °C (Appendix A). The total weight fraction of the dicyclopentyl and dimethyl groups, as estimated from the NMR-derived composition (35.9 wt%) is in good agreement with the experimental weight loss measured by TGA (34.4 wt%) (Appendix A).

### 2.4. Morphological and Porosity Analysis of PSS-3 and PSS-4

The morphology of PSS materials was evaluated by means of scanning electron microscopy (SEM). Large particles with random morphology were observed for PSS-3 as well as PSS-4 (Figure 5). Some superficial porosity, in the form of macropores, might be observed but are rather rare. As expected, PSS-3 nor PSS-4 could not be characterized by N_2_ physisorption. N_2_ physisorption typically fails for PSS materials because the temperature conditions at which N_2_ physisorption occurs cause the flexible matrix to collapse, a phenomenon also observed in other silicate hybrid materials with flexible linkers [8,28,30].

To gain some insight into the presence or absence of porosity, HAADF-STEM tomography was performed on PSS-4 (Figure 5c). This analysis indicated a porosity of 28%, mainly caused by a broad pore size distribution comprising meso- and macropores.

## 3. Materials and Methods

### 3.1. Chemical and Reagents

TEOS (Acros organics, Geel, Belgium, 98%), ammonium hydroxide (Chem-lab, Zedelgem, Belgium, 29wt% in water), tetrabutylammonium hydroxide (TBA-OH) (Acros organics, 40 wt% in water), tetrahydrofuran (THF) (Acros organics, 99.8%, extra dry over molecular sieve, stabilized, acrosealTM), dimethyldichlorosilane (Acros organics, >99%, acroseal), octylmethyldichlorosilane (Alfa Aesar, Ward Hill, MA, USA 98%), dicyclopentyldichlorosilane (Gelest, Morrisville, PA, USA, 97%), and potassium sulfate (Acros organics, ≥ 99%) were used as purchased, without any further purification.

### 3.2. Tetrabutylammonium Cyclosilicate Hydrate (TBA-CySH) Synthesis

Tetrabutylammonium cyclosilicate hydrate synthesis is based on the work by Smet et al. (2017) [17]. TEOS (139 mL) was added dropwise to a stirred solution of tetrabutylammonium—(140.52 mL) and ammonium hydroxide (222.47 mL) [1TBAOH:7.8NH_4_OH:59H_2_O on a molar basis], to prevent gelation. After 48 h, a white suspension was filtered (Whatman^™^ qualitative filter paper grade 5; 90 mm diameter, Cytiva, Marlborough, MA, USA) and washed with water. After drying in ambient conditions, TBA-CySH crystals were recovered.

### 3.3. PSS-3 Synthesis

TBA-CySH (1 g) was dried under a vacuum (1 mbar) at room temperature for 72 h in a 100 mL, 2-neck round-bottom flask, sealed with rubber stoppers, and a high-vacuum silicone grease connected to a standard Schlenk line. THF (12 mL) and octylmethyldichlorosilane were added to the pre-dried TBA-CySH crystals. Respectively, 1.3 mL, 2.6 mL, or 3.9 mL of octylmethyldichlorosilane were added to achieve a 50% deficit, stoichiometric ratio, and 50% excess PSS-3. After 6 h of reaction, THF and excessive silane were evaporated under a vacuum (1 mbar) at 90 °C for 16 h to yield an intermediate spherosilicate powder. To generate PSS-3, these spherosilicates were dissolved in THF (14 mL) and subsequently reacted with water vapor in a desiccator at 98% relative humidity. After 24 h, PSS-3 powders were collected and dried at 60 °C at ambient pressure.

### 3.4. PSS-4 Synthesis

PSS-4 was synthesized via a similar procedure as PSS-3, but other silanes were added. Dicyclopentyl- (0.74 mL) and dimethyldichlorosilane (1.1 mL) were mixed together in a glass vial under an inert atmosphere prior to the addition to pre-dried TBA-CySH dissolved in THF (12 mL). The molar ratio of CySH, dicyclopentyl- and dimethyldichlorosilane was 1/3/8. After 6 h of reaction, THF and excessive silane were evaporated at vacuum and room temperature for 16 h to yield an intermediate spherosilicate powder. To yield to PSS-4, these spherosilicates were redissolved in THF (14 mL) and reacted to with water vapor in a desiccator at 98% relative humidity. After 24 h, PSS-4 powders were collected and dried at 60 °C at an ambient pressure.

### 3.5. Calcination of PSS Materials

PSS-3 was calcined in a U-tube flow oven under an N_2_ atmosphere at 250 °C with a rate of 1 °C/min. The calcination temperature was kept for 2 h before gradually being cooled to room temperature. PSS-4 was calcined via a similar procedure but at 200 °C for 4 h.

### 3.6. Characterization of PSS Materials

Powder XRD patterns were recorded in transmission mode on a high-throughput STOE stadi P diffractometer (STOE & Cie GmbH, Darmstadt, Germany) equipped with an image plate detector. Thermogravimetric analyses under the N_2_ atmosphere were performed using a TGA/DSC^3+^ apparatus (Mettler Toledo, Greifensee, Switzerland). At a rate of 10 °C/min, each sample was heated from room temperature to 900 °C before gradually being cooled back to ambient conditions. The temperature at which a weight loss of 5 wt% was observed is denoted as T_5%_ and is referred to as the thermal stability temperature of the POSiSil materials. Scanning electron microscopy images were taken with a Nova NanoSEM 450 (FEI, Eindhoven, The Netherlands) at a voltage of 2 kV. The powders obtained upon crushing the materials were dispersed on carbon tape and imaged as such, without additional coating. High-Angle Annular Dark-Field Scanning Transmission Electron Microscopy (HAADF-STEM) of images at room temperature (RT) has been performed on an aberration-corrected cubed Thermo Fisher Titan microscope (Thermo Fischer Scientific, Waltham, MA, USA) operating at 300 kV. Electron tomography tilt series were acquired using a Fischione 2020 tomography holder (E.A. Fischione Instruments, Inc., Export, PA, USA). A high-resolution HAADF-STEM tilt series was acquired over a tilt range between ±75° with a tilt increment of 3°. To obtain a reliable reconstruction of the pore structure, a recently developed approach was used for which a time series of images was acquired for every tilt angle at a short dwell time [48]. The projection images were acquired using 5 frames at 1 µs dwell time, while the image resolution was set to 1024 × 1024 pixels. Next, these images were used as inputs for a non-rigid registration method in combination with a convolutional neural network (CNN) [48]. The obtained series were aligned using cross-correlation, and 3D reconstructions were obtained using the Expectation Maximization (EM) algorithm implemented in Astra Toolbox (v2.1) [49,50].

All solid-state NMR experiments were carried out in Bruker 300 Avance III (7.4 T) and Bruker 500 Avance III (11.4 T) spectrometers (Bruker BioSpin, Ettlingen, Germany) with respective ^29^Si Larmor frequencies of 59.63 MHz and 99.36 MHz, with 4 mm triple resonance MAS probes. For PSS-3, the octyl-methyl silicone polymer and the stoichiometric PSS-3 sample were recorded at 300 MHz with a ^1^H decoupled direct excitation sequence with 320 and 304 transients, respectively, and a recycle delay of 150 and 240 s, respectively, using a π/2 pulse of radiofrequency strength of 66 kHz and a ^1^H decoupling (strength of ~55 kHz) achieved using the SW_f_-SPINAL sequence [51]. For the 50% deficit and excess PSS-3 samples, ^1^H decoupled ^29^Si MAS spectra were recorded at 500 MHz with 240 and 128 transients, respectively, and a recycle delay of 300 s, using a π/2 pulse of radiofrequency strength of 60 kHz and ^1^H decoupling (strength of ~55 kHz) achieved using a SW_f_-SPINAL sequence [51]. For PSS-4, all measurements were performed at 500 MHz using the above sequence and parameters. In total, 136 transients were recorded with a recycle delay of 400 s. The ^1^H NMR experiments were performed with 8 scans and a 15 s recycle delay. The ^29^Si chemical shifts were referenced against the resonances of Q_8_M_8_, a secondary reference of tetramethylsilane (TMS). The ^1^H-^1^H double quantum-single quantum (DQ-SQ) experiments were performed with BABA pulses of 50 kHz strength [52]. In total, 256 slices were collected with an increment of 66.67 µs (rotor synchronized). The ^29^Si CP-RFDR experiment was conducted at 20 kHz MAS using a CP contact of 6.5 ms and ^29^Si pulses of 70 kHz strength. In total, 700 slices were collected with an increment of 66.67 µs (rotor synchronized), with 416 transients recorded for each slice at an RFDR mixing time of 90 ms. The 2D ^1^H-^13^C CP-HETCOR experiment was carried out using a CP contact time of 0.5 ms. A recycle delay of 4 s was used for 32 scans per slice of the 2D experiment. With an increment of 19 µs, 640 slices were recorded. The 2D ^1^H-^29^Si CP-HETCOR experiment was carried out using a CP contact time of 2 ms. A recycle delay of 5 s was used for 176 scans per slice of the 2D experiment. With an increment of 25 µs, 256 slices were recorded. All NMR spectra fittings were performed by using the Dmfit software (v. #20200306) [53].

## 4. Conclusions

In summary, this research demonstrated the successful incorporation of long and bulky hydrocarbon functionalities in novel silicate-silicone hybrid materials, belonging to the PolyOligoSiloxySilicone (POSiSil) material class. Both PSS-3 and PSS-4 syntheses started from zero-dimensional hydrogen-bonded tetrabutylammonium cyclosilicate hydrate crystals constituted of silicate cubes [Si_8_O_20_], a silicate precursor enclosing a low amount of crystal water that allows using organodichlorosilane reagents as linkers. Octyl-methyl and a mix of dimethyl—and dicyclopentylsilicone moieties were incorporated in PSS-3 and PSS-4, respectively, through silylation of silicate cubes with the corresponding dichlorosilanes, followed by subsequent hydrolysis with water (vapor). The linear, aliphatic octyl hydrocarbon in PSS-3 induced a maximum average length of the silicone linkers of two. Steric hindrance was experienced with dicyclopentyl functionalities, which was bypassed using an additional, “smaller” dimethyldichlorosilane reagent. A mixed dimethyl-dicyclopentyl silicone linker, (Si_8_-O-Si(Me_2_)-O-Si(Cp_2_)-O-Si_8_), was identified. Multidimensional ^1^H, ^13^C, and ^29^Si NMR spectroscopy were employed for the quantitative characterization of silicone linkers in PSS-3 and PSS-4, addressing a scientific gap in previous studies on similar materials with large hydrocarbon functionalities. Physical properties like thermal stability up to 400 °C were observed for amorphous PSS particles. Looking at the overall porosity with HAADF-STEM, PSS-4 with the bulkier cyclopentyl functionalities showed a porosity of 28%. A detailed characterization of the porosity of PSS-3 and PSS-4 is necessary, but the collapse of the sample’s flexible network at 77 K renders conventional N_2_ physisorption unsuitable. Water intrusion-extrusion cycles could be a potential alternative.

The controlled incorporation of large hydrocarbon functionalities could facilitate a mesostructured or hierarchical pore network due to their compatibility with reverse micelles. Hydrophobic interactions between reverse micelle tails and, for example, the linear octyl moieties of PSS-3 could organize the silicate-silicone hybrid material around the micellar shape or even self-assembly into an ordered mesoscopic fashion, a feature only achieved by few other reports, albeit suffering from a low synthesis efficiency starting from POSS precursors [54,55,56]. The unique set of properties of silicate-silicone hybrid materials, combined with in situ chemical functionalization, offer potential applications such as high-pressure gas storage, catalysis, and adsorption and separation processes.

## Figures and Tables

**Figure 1 molecules-30-00228-f001:**
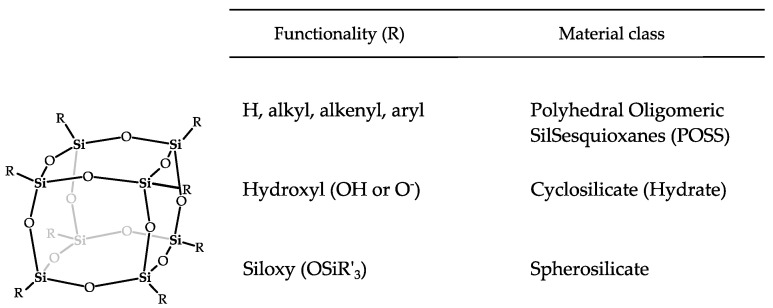
Chemical structure of the silicate cube (D4R) (**left**) and specification of peripheral chemical functionality (R) differentiating the material classes containing this building block (**right**).

**Figure 2 molecules-30-00228-f002:**
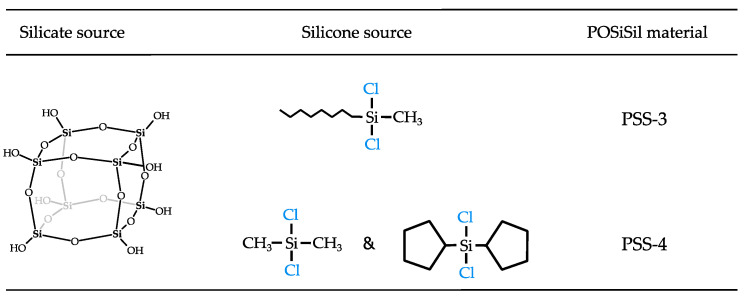
Overview of new POSiSil materials presented in this work, PSS-3 and PSS-4, with their respective silicate and silicone composition. Silicate cubes are sourced from a TBA-CySH crystal.

**Figure 3 molecules-30-00228-f003:**
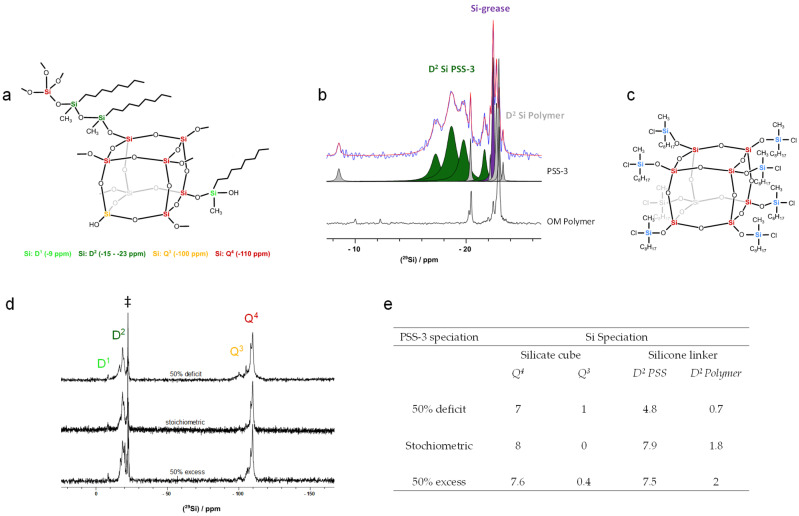
(**a**) The chemical structure model of different silicon species present in an octyl-methyl POSiSil with their NMR chemical shift. (**b**) ^1^H decoupled ^29^Si single-pulse MAS NMR spectra of an octyl-methyl POSiSil (50% excess, top) and the silicone polymer of n-octyl(methyl)dichlorosilane (OM polymer, bottom), focusing on D-coordinated Si atoms. The decomposition of the PSS spectrum is given, and the colors are ascribed to the phase they occur, i.e., the PSS-3 network (green), silicone polymer fraction (gray), and silicone grease (purple). (**c**) The chemical structure model of spherosilicate with monomeric octyl-methyl silyl moieties. (**d**) ^1^H decoupled ^29^Si direct excitation MAS NMR spectra of PSS-3 materials with different concentrations of n-octyl(methyl)dichlorosilane. The sharp resonance (‡) at −22.4 ppm is assigned to contamination with silicone grease, which was applied to the ground glass joints of the reaction set-up, also covering some silicone polymer signals. Better visualization is provided in (**b**). (**e**) Quantitative numbers of Q^4^, Q^3^, and D^2^-coordinated ^29^Si atoms per silicate cube of PSS-3, based on Appendix A, according to the different silane concentrations applied in the synthesis protocol. D^2^ Si atoms were differentiated, belonging to either the PSS or the polymer phase.

**Figure 4 molecules-30-00228-f004:**
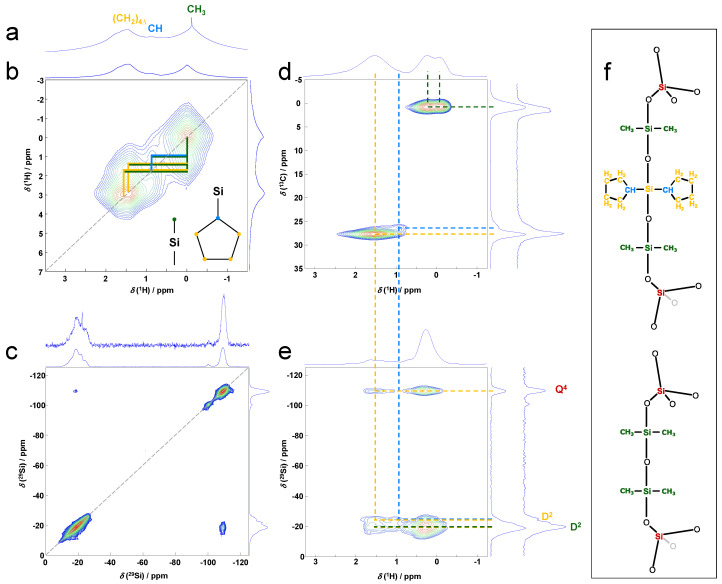
Solid-state NMR crystallography analysis of PSS-4. (**a**) 1D direct excitation ^1^H MAS NMR spectrum; (**b**) 2D ^1^H-^1^H dipolar DQ-SQ correlation spectrum. Cross-correlations between the protons of the dimethylsilyl and dicyclopentyl groups are marked with full lines in the colors used to mark the respective protons in the structure model in (**f**); (**c**) 1D ^1^H decoupled ^29^Si single-pulse MAS NMR spectrum with their corresponding 2D ^29^Si-^29^Si CP RFDR spectrum. ^29^Si atoms of pure dimethylsilicone linkers correlate with the silicate cube, represented by the cross-correlations (off-diagonal); (**d**) 2D ^1^H-^13^C HETCOR spectrum; (**e**) 2D ^1^H-^29^Si HETCOR spectrum. The correlations are indicated with dashed lines corresponding to the colors of the ^1^H and ^13^C atoms in the structure model in (**f**). (**f**) The chemical structure model with a mixed silicone linker (-O-SiMe_2_-O-SiCp_2_-O-) and a pure dimethylsilicone linker (-O-(SiMe_2_-O)_3_-).

**Figure 5 molecules-30-00228-f005:**
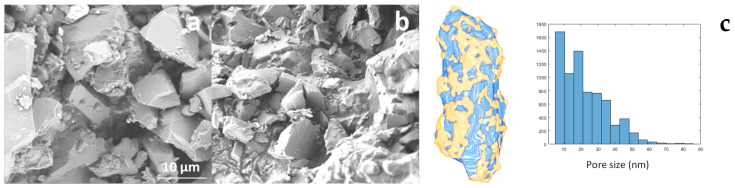
SEM images of PSS-3 with a 50% deficit of silane (**a**) and PSS-4 (**b**). HAADF-STEM tomography and data analysis of PSS-4. (**c**) The data illustrate the 3D volume, showing the pore volume (blue), material (yellow) (**left**), and pore size distribution (**right**).

## Data Availability

Data are contained within the article and Appendix A.

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
