# Peer review of "Synthesis and Advanced NMR Characterization of Ordered 3D Reticular Materials with PolySilicate Nodes and Hydrophobic OrganoSilicone Linkers"

_molecules, 2025, doi:10.3390/molecules30020228_

Round 1

Reviewer 1 Report

Comments and Suggestions for Authors

The work reported in this manuscript brings together two important areas of silicon chemistry, those of polyhedral oligosilsesquioxanes (POSS) and silicones. Linking POSS cubes via siloxane chains bearing organic groups larger than methyl, in this case octyl and cyclopentyl, by including the groups in the silanes used to react with a POSS reagent is novel and gives interesting materials which will, no doubt be the basis for much future work. The synthesis procedures used are clearly described and the methods used for materials analysis are appropriate. The conclusions drawn appear to be well founded. I think that the pape4r can be accepted for publication with only a few small points that should be addressed beforehand, as follow:

1.      Abstract, would read better as “allowed tuning of the length …”

2.      Line 140, better as “allows differentiation between …”

3.      Line 142, needs “respectively” prior to (Figure 3a)

4.      Line 143 and elsewhere , spectra are described as decomposed or subject to decomposition. Better would be to use the more scientific word  ‘deconvolution’ for this process.

5.      Line 154, ‘moieties’ is incorrect in this case. Moiety means a large part of something, about half. An Si-Cl group should be called a group, not a moiety.

6.      Line 154 better as “react with a free silanol”

7.      Line 188 and elsewhere, why do the Si chemists use silicone grease when they know it interferes with the 29Si NMR spectra which are so important in this work. Better would be to use a hydrocarbon grease or Teflon sleeve.

8.      Line 221, better as “not allow formation of…”

9.      Line 223 and below. Defects are mentioned but it is not clear what these defects are, some clarification needs to be added here.

10.  Line 229, a ration of reagents of 8 to 3 is mentioned, why is this ratio used? What is its significance?

11.  Line 324, crystals are mentioned. Is this material crystalline or amorphous?

12.  3.3 and 3.4, materials are dried at 60C, but at what pressure?

13.  Line 401 says that hydrophobization has been demonstrated but this is not strictly true. To demonstrate hydrophobization some measurements of water uptake or repellency would be needed, and these have not been done. What has been demonstrated is that POSS cubes can be linked by siloxane groups bearing large, hydrophobic organic substituents.  Demonstrating hydrophobicity will, no doubt, come in a later publication.  This line of the conclusion needs to be re-worded to make it clearer.

14.  Figure S6, what does the red component to the spectrum correlate with in the structure – there are no red labels of protons?

Author Response

kindly refer to the pdf file attached

Reviewer 2 Report

Comments and Suggestions for Authors

The work by Breynaert reports the synthesis and characterization of cage silsesquioxanes and organic silane linkers. Overall, the experiment data are convincing and interesting. Conceptually, this work is innovative and has the potential to significantly impact the materials field. This reviewer recommends accepting a revised version of this manuscript for publication, provided the following comments are addressed:

1. The table in Figure 3e is too simple and omits important information. For the PSS-3 spectra (Figures 3 and S3), the authors should provide the assignment and the corresponding intensity for each resonance after deconvolution (including dimeric, monomeric, or trimeric D2 sites, and different Q4 and Q3 sites). Without this information, it is too hard to understand the discussion in the manuscript quantitatively.

2. In Figure S3, the spectra show higher Q3 intensity in reactions with excess dichlorosilane. It may indicate that some dichlorosilane can form Q3 sites through self-polymerization. However, the authors did not discuss this observation in the manuscript.

3. In line 201, the authors imply that "the length variation of the silicone linker also explains multiple Q4 Si signals of silicate cubes in the 29Si NMR spectrum." Do the authors have references to support this explanation?

4. In the spectrum shown in Figure S3, there are multiple Q4 signals between -105 ppm and -110 ppm. It is difficult to understand why the Q4 signals in POSS are significantly affected by the variation in bridge linker length (from one to three units). The authors could consider discussing the chemical shift distribution of Q4 in the context of the previous review: *Journal of Sol-Gel Science and Technology* (2022) 104:36–52.

5. In Supporting Table S1, the authors assign both the -13 ppm and -24.8 ppm signals to the D2-Si(Cp)2 site. However, the 1H-29Si HETCOR does not have sufficient resolution in the proton dimension to support this assignment. 

6. The 1H spectra acquired with MAS at 15 kHz are not entirely convincing, especially due to the differing linewidths of the CH3 group signals. Could the authors consider running the 1H NMR experiments at a higher spinning rate (e.g., >40 kHz) to achieve higher resolution in the 1H NMR spectra?

Author Response

kindly refer to the pdf file attached

Round 2

Reviewer 2 Report

Comments and Suggestions for Authors

The authors have addressed most of the aspects requested by the reviewers. Fast spinning ¹H NMR data have been added. The intensities of each D and Q site have also been included in the supporting information. Additional references have been incorporated. Overall, the experimental data are more convincing and complete. This reviewer recommends accepting the revised version of this manuscript